# Process Optimization for the 3D Printing of PLA and HNT Composites with Arburg Plastic Freeforming

Leonardo G. Engler [1,2], Janaina S. Crespo [1,2], Noel M. Gately [3], Ian Major [1] and Declan M. Devine [1,*]

1. PRISM Research Institute, Technological University of the Shannon: Midlands Midwest, Athlone Campus, University Road, N37 HD68 Athlone, Ireland
2. Postgraduate Program in Materials Science and Engineering, University of Caxias do Sul, Francisco Getúlio Vargas Street, 1130, Caxias do Sul 95070-560, Brazil
3. Applied Polymer Technologies Gateway, Technological University of the Shannon: Midlands Midwest, Athlone Campus, University Road, N37 HD68 Athlone, Ireland
* Correspondence: ddevine@ait.ie

**Abstract:** The industrial use of additive manufacturing continues to rapidly increase as new technology developments become available. The Arburg plastic freeforming (APF) process is designed to utilize standard polymeric granules in order to print parts with properties similar to those of molded parts. Despite the emerging industrial importance of APF, the current body of knowledge regarding this technology is still very limited, especially in the field of biodegradable polymer composites. To this end, poly(lactic acid) (PLA) was reinforced with halloysite nanotubes (HNTs) by hot melt extrusion. The PLA/HNT (0–10 wt%.) composites were analyzed in terms of their rheology, morphology, and thermal and mechanical properties. A study of the processing properties of these composites in the context of APF was performed to ensure the consistency of 3D-printed, high-quality components. The optimized machine settings were used to evaluate the tensile properties of specimens printed with different axis orientations (XY and XZ) and deposition angles (0 and 45°). Specimens printed with an XY orientation and deposition angle starting at 0° resulted in the highest mechanical properties. In this study, the use of PLA/HNT composites in an APF process was reported for the first time, and the current methodology achieved satisfactory results in terms of the 3D printing and evaluation of successful PLA/HNT composites to be used as feedstock in an APF process.

**Keywords:** Arburg plastic freeforming (APF); poly(lactic acid) (PLA); halloysite nanotubes (HNTs); mechanical properties; APF process parameters; 3D printing; additive manufacturing; biodegradable composites

## 1. Introduction

Biodegradable polymers have become increasingly popular worldwide in recent years. However, biodegradable polymers have some limitations on their use due to their price and thermostability, as well as their mechanical and physical properties. Poly(lactic acid) (PLA) is a biodegradable polyester that has been employed in a variety of applications, including packaging, agriculture, and medicine, due to its specific properties, such as a good biodegradability, biocompatibility, and mechanical performance, in addition to a high strength and modulus resistance [1]. PLA and HNT composites were selected as the materials for the study because PLA is an inert, biocompatible, transparent, and resistant thermoplastic, while HNTs can improve the mechanical properties of the polymeric matrix and, due to their hollow-tube structure, are able to entrap different active agents, such as drugs or proteins. PLA has been utilized in medical devices for the implantation of prostheses and other medical products for decades because of these particular properties. Nevertheless, the addition of HNTs can compensate for some limitations of PLA, such as its low thermal stability and poor barrier properties [2,3]. In terms of biomedical applications, it has been reported that PLA combined with halloysite nanotubes (HNT) can

be advantageous [4–7]; however, due to printing technology constraints, the 3D printing of these composites has proven difficult [1].

The use of 3D printing or additive manufacturing (AM) is trending as a method of fast prototyping, particularly in some specific industries, such as medical device manufacturing. AM enables the use of patient-specific medicine and the production of medical equipment with complicated designs that would be prohibitively expensive to produce using conventional technologies [8].

Since the Arburg plastic freeforming (APF) process is a relatively new additive manufacturing technology, there is a lack of literature available based on different polymers and different combinations of materials. In the APF process, there are two main characteristics that have the most prominent effects on the mechanical performance of samples printed by droplet deposition, and therefore, these have been more carefully investigated in the literature. Firstly, the processing parameters selected have a significant impact on the final quality of the printed components. APF involves a relatively complex mechanism based on the interaction of different parameters, such as the temperature of both the melt and build chamber, the drop aspect ratio (DAR), which is the ratio between the width and the height of a deposited droplet of material and is influenced by the rheology of the material, and the discharge rate. To produce high-performing components, these parameters must be optimized, and their interaction with one another and with the properties of the printed components must be investigated [8–14]. Secondly, given that the sample building orientation has a major impact on the final mechanical properties, some authors focused their studies on this topic [15,16]. Other APF parameters have been only partially investigated in the literature, as the number of tests used to assess all the parameters could not feasibly be described in a single work [8–14]. In the work of Zhang et al. [10], the APF was used to manufacture tablets with different infill percentages using the pharmaceutical polymers hypromellose acetate succinate (HPMCAS) and polyethylene oxide (PEO). The authors discovered, for the first time, that there is a linear correlation between the drug release rate and infill percentage when the pore size is not significantly affected by swelling, suggesting that the porosity of the parts produced by APF can be used to control the drug release rate in swellable and erodible systems. In another study by Charlon and Soulestin [17] using acrylonitrile butadiene styrene (ABS) as a feedstock material, the authors printed over 400 dumbbell-shaped samples to evaluate many different mechanical properties, assessing the discharge parameter, filling density, number of contour lines, and overlapping, as well as some process parameters, such as the nozzle and build chamber temperature and the processing speed and angle. They concluded the APF represents a great advantage compared to traditional 3D printers, as it uses polymeric pellets as feedstock instead of filaments. However, the high number of adjustable parameters complicates the utilization of the process. Hence, in order to achieve samples with good mechanical properties, it is recommended to utilize a high nozzle temperature, discharge rate, and filling density, while the slicing distance should be low so as to promote the raster cohesion between printed layers. Some studies of the degradation of the medical-grade copolymers, poly(L-lactide-co-ε-caprolactone) (PCLA) and poly(L-lactide-co-trimethylene carbonate) (PLATMC), have shown that it is necessary to identify a processing window considering the thermal and mechanical properties of the materials and regarding the manufacturing of medical devices and scaffolds, as this might affect its immediate performance and its long-term degradation behavior [14].

However, to the best of our knowledge, there is no published literature on the study of the APF process parameters of PLA-based composites and how these parameters affect the final characteristics of printed parts. However, this study aims to qualify and investigate the properties of composites of PLA and HNT with respect to their potential use as an additive manufacturing feedstock for 3D printing (namely, Arburg Freeformer 300-3X (ARBURG GmbH + Co KG, Lossburg, Germany)).

## 2. Materials and Methods

Polylactic acid (PLA) was supplied from Corbion PLA LX 175 in pellet form. It had a density of 1.24 g·cm$^{-3}$, a minimum L-isomer content of 99%, a melting temperature of 175 °C, a glass transition temperature of 55 °C, a tensile modulus of 3500 MPa, and a strength of 45 MPa. Sigma-Aldrich Ireland provided the Dragonite-HP halloysite nanotubes (HNTs) in powder form. They had a density of 2.56 g·cm$^{-3}$, a cylindrical form with a length of 0.2–2 µm, an outer diameter of 50–70 nm, an inner diameter of 15–45 nm, and a length-to-diameter ratio of 10–20.

### 2.1. Processing

The material compounding was performed using a Prism TSE 16 (Thermo Electron, Staffordshire, UK) twin-screw extruder with 16 mm-diameter screws and a 25/1 length to diameter ratio (L/D). The profile temperature used was kept at 200/190/180/150/100 °C from the die to the feeder. The screw rate was maintained as constant at 135 rpm during the extrusion of the compounding of the PLA/HNT in the following mass fractions of HNT: 0, 1, 3, 5, and 10% wt. These compounds are referred to in this study as PLA, PLA/HNT1, PLA/HNT3, PLA/HNT5, and PLA/HNT10, respectively. The extrudate was solidified using an air-cooled conveyor belt and pelletized at a nominal length of 2 mm.

### 2.2. MFI and Rheological Studies

To assess the melt flow properties of the polymers, the melt flow indexing was measured using a Zwick Rowell Cflow extrusion plastometer. The temperature was set to 190 °C and, following the ISO 1133 [18], a standard weight of 2.16 kg was used for the test, carried out using a die with a nominal length of 8.00 mm and a bore diameter of 2.095 mm. Samples were cut every 30 s and the weight of each cut was measured using a laboratory scale. The results for MFI are presented as grams per ten minutes as standard. To support the MFI results, a rheological analysis of all the samples was performed using an oscillatory rheometer, the TA Discovery Hybrid Rheometer HR30 (New Castle, PA, USA), with a 25 mm parallel plate geometry. A 1% amplitude was used, and it was previously validated by an amplitude sweep at a frequency of 1.0 Hz. At a constant temperature of 190 °C, oscillation frequency sweeps from a 0.1 to 100 rad/s angular frequency were performed.

### 2.3. Differential Scanning Calorimetry (DSC)

DSC tests were carried out on samples weighing between 8 and 12 mg. The samples were crimped in non-perforated aluminum pans with reference to an empty crimped aluminum pan. The DSC analysis was performed using a Pyris 6 DSC (Perkin Elmer, MA, USA) with a nitrogen flow rate of 30 mL·min$^{-1}$ to avoid material oxidation. To eliminate the previous thermal history of the material, the samples were heated from 20 to 220 °C at the rate of 10 °C·min$^{-1}$ and then held isothermally at 220 °C for 5 min. Then, the samples were cooled from 220 to 20 °C at 10 °C·min$^{-1}$. Lastly, the thermal properties of the samples were recorded by heating the samples again from 20 to 220 °C at the rate of 10 °C·min$^{-1}$, and the glass transition temperature, cold crystallization temperature, and melting temperature of each sample were recorded [1].

### 2.4. Thermogravimetric Analysis (TGA)

The TGA was performed to evaluate the thermal stability of the compounded PLA/HNT composites using a thermal analyzer (TGA-50, Shimadzu Co., Kyoto, Japan) with a dynamic nitrogen atmosphere of 50 mL·min$^{-1}$. Samples weighing between 8–12 mg were heated from room temperature (20 °C) to 900 °C at a heating rate of 10 °C·min$^{-1}$, and the weight loss of the samples was plotted against the temperature.

### 2.5. Fourier Transform Infrared Spectroscopy Analysis (FTIR)

A Perkin Elmer Spectrum One (PerkinElmer, Waltham, MA, USA) with a universal attenuated total reflectance (ATR) sample adapter was used for the FTIR analysis to detect

the interaction level between the HNT and the PLA matrix compounding. All data were collected at the room temperature of 21 °C in the spectral range between 4000–650 cm$^{-1}$ against a background of air, using 4 scans per sample cycle with a resolution of 1 cm$^{-1}$ and a constant universal compression force of 80 N. Following that, spectrum software was used to perform the analysis.

### 2.6. Wide-Angle X-Ray Diffraction (WAXD) Analysis

Wide-angle X-ray diffraction (WAXD) was used to identify the crystalline phases present in the material and ways in which the addition of HNT to the polymeric matrix affected their crystallinity. Thin films were prepared by compression molding at 160 °C for 3 min with 2 tons pressure. The sample preparation consisted of the removal of a rectangular sample of the polymeric thin films and its subsequent fixation on a glass plate. The experiments were performed with a Shimadzu XRD-6000 diffractometer using Cu K$\alpha$ radiation ($\lambda$ = 1.5405 Å) in the reflection mode. Data were collected over 2$\theta$ angles ranging from 2 to 40°, with a step of 0.05° s$^{-1}$ and an integration time of 2 s. Bragg's law was used to calculate the d-spacing between the crystal lattice planes of atoms that produced constructive interference between different planes. The Bragg's law equation is described below:

$$n \lambda = 2d \sin \theta \tag{1}$$

where n is an integer and represents the reflection order or the diffraction. In this case, its value is equal to 2, while $\lambda$ is the wavelength (in nm), d is the interplanar spacing (in nm), and $\theta$ is the angle of incidence.

### 2.7. Part Production

An Arburg Freeformer 300-3X (ARBURG GmbH + Co KG, Lossburg, Germany) additive manufacturing machine was used to fabricate the specimens. A schematic representation of the Arburg plastic freeforming process is shown and described by Pollack et al. [19]. The 3D samples were designed using the CAD software SolidWorks (Dassault Systèmes, Velizy-Villacoublay, France), and STL (standard triangle language) files were generated and loaded into the Arburg Freeformer slicing software v2.30 (ARBURG GmbH + Co KG, Lossburg, Germany). The diameter of the nozzle used was 0.2 mm. Further information regarding the process parameters used for the 3D printing of the parts is presented in Table 1.

**Table 1.** Arburg Freeformer process parameters for PLA and PLA/HNT5.

| Parameter | PLA | PLA/HNT5 |
|---|---|---|
| Temperature zone 1 ($T_1$, °C) | 210 | 215 |
| Temperature zone 2 ($T_2$, °C) | 220 | 225 |
| Temperature nozzle ($T_{nozzle}$, °C) | 225 | 230 |
| Temperature chamber ($T_{chamber}$, °C) | 90 | 90 |
| Dosing stroke (mm) | 8 | 8 |
| Backpressure (bar) | 50 | 50 |
| Screw speed (m/s) | 4 | 4 |
| Discharge rate (%) | 100 | 100 |
| Drop aspect ratio (DAR) | 1.52 | 1.58 |
| Layer thickness (mm) | 0.200 | 0.200 |

Prior to printing, the material was dried for 24 h at 80 ± 1 °C to ensure a moisture content of less than 0.04%, as per the processing recommendations. The moisture content was measured using the MB90 moisture analyzer (OHAUS Europe GmbH, Heuwinkelstrasse 3, Switzerland), and we verified a moisture content of 0.022 ± 0.01% after drying [20]. The choice of the drop aspect ratio (DAR), which is the ratio between the nominal droplet width and the layer height, defines the distance between the center of the droplets and is defined by the computer-aided manufacturing tool before slicing, although the ideal

DAR value is determined experimentally. The rheological properties of the material at a certain temperature and processing pressure are used to define the real droplet width [13]. Cubes of PLA and PLA/HNT5 were printed with varied DAR values to test the printing quality with different DAR values, as seen in Figure 1a,b. The porosity of these cubes was assessed to verify the real infill percentage of the samples after printing. After this infill was optimized, and the DAR value was used to print samples for the mechanical tests.

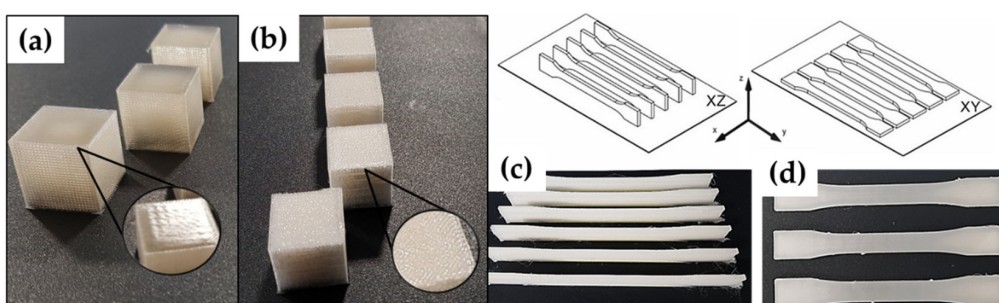

**Figure 1.** Cubes printed with various DAR values. Desirable glossy final part quality in (**a**) and under/overfilled opaque part quality in (**b**). PLA tensile specimens were printed with DAR values of 1.52 in different axis orientations: XZ in (**c**) and XY in (**d**).

After defining the DAR values, tensile samples were printed to assess the mechanical properties of the printed parts, as shown in Figure 1c,d. PLA tensile specimens with a DAR value of 1.52, using different axis orientations and varying deposition angles, were printed. The axis orientations XY and XZ were used, and the deposition angle was varied by printing the first layer in the 0° and 45° directions, and each subsequent layer was deposited in a position 90° higher than the preceding layer.

### 2.8. Infill vs. Porosity Assessment

The relationship between the infill and porosity of the printed parts was assessed using printing cubes of the same dimensions. Infill versus porosity tests were performed to assess the experimental densities of the printed parts with different DARs, and we compared them to the theoretical values. The porosity measurements were performed at room temperature (20 °C) following Archimedes' principle of measurement of the density of solids, according to the method described in the literature [21]. First, the weights of the dry samples were measured following submergence in water, and subsequent repeated weight measurements were performed applying the correction factors for the density of the air and the water at 20 °C and the density of the materials based on the supplier's datasheets. Thus, it was possible to determine the interplay between the DAR values and the porosity of the printed components. Cubes with a 100-layer height (20 × 20 × 20 mm) were printed with different DAR values to verify the visual appearance of the cubes, as well as the relation between the porosity and infill percentage of the cubes, calculated experimentally. These tests were performed by varying the DAR values and identifying underfilled and overfilled regions in the samples, with an emphasis on the identification of parts that had a glossy appearance and were as close as possible to the infill at 100% (experimental value). DAR values ranging between 1.40 and 1.75 were used when printing the cubes of PLA and PLA/HNT5.

### 2.9. Tensile Testing

Tensile testing was performed in accordance with the EN ISO 527 standard for test specimens using a ZwickRoell Z010 universal testing machine (ZwickRoell Ltd., Ulm, Germany) and a 5 kN load cell. A total of 6 specimens for each sample were tested, with dimensions of 170 mm in length, a width of 10 ± 0.2 mm, and a thickness of 4 ± 0.2 mm. The tests were carried out at a strain rate of 5 mm·min$^{-1}$ at room temperature, with a pull-to-limit setup (stop at a strain of 20% from the maximum strain). The tensile properties

were obtained directly from the machine. A minimum of five replicates were evaluated per group, and the exact dimensions of each sample were recorded prior to testing. The stress–strain curves for each sample were recorded using the software TestXpert II (ZwickRoell, Ulm, Germany) and utilized for the calculation of the Young Modulus.

### 2.10. Scanning Electron Microscopy (SEM)

The samples were analyzed using field emission scanning electron microscopy (FE-SEM) to gather information on the morphology and size of the HNT nanotubes in the PLA matrix and to assess the deposited distribution of droplets on the samples. The FESEM studies were performed in a Mira 3 TESCAN (10 kV). To observe the internal layers of the printed parts, cryogenic fracture was performed on these samples using liquid nitrogen. Prior to analysis, the samples were deposited on appropriate metallic stubs and sputter-coated with gold to increase the conductivity of the materials under investigation.

## 3. Results and Discussion

### 3.1. Hot-Melt Extrusion (HME)

The hot-melt extrusion of the PLA-based composites was successful, with no observable degradation of the compounded extrudate. The twin-screw extruder employed in this processing step was equipped with a set of modular twin screws with three kneading areas, which increases the dispersive mixing properties of the process [22,23]. The shorter barrel of the extruder contributes to the reduction in the residence time of the material, which decreases the exposure of the thermally sensitive compounds to heat and shear stress, preventing their early degradation [23,24].

### 3.2. Melt Flow Indexing and Rheological Studies

The MFI data showed that the addition of HNT caused a reduction in the MFI values, implying an increase in the material's viscosity, as seen in Figure 2a. This phenomenon is likely to be due to the exfoliation/intercalation shear of the nanofiller during testing [25,26].

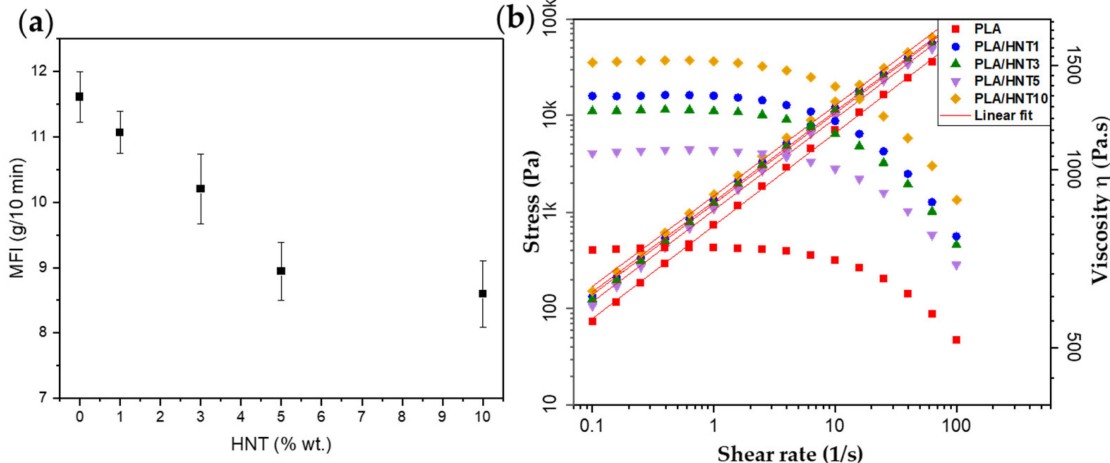

**Figure 2.** Melt flow index of PLA and PLA/HNT samples at 190 °C in (**a**). Viscosity η (Pa·s) and stress σ (MPa) by shear rate $\dot{\gamma}$ (1/s) in (**b**). Data corrected using the Cox–Merz rule.

Figure 2b shows the viscosity and stress by the shear rate of the PLA and PLA/HNT composites. The relationship between the stress rate and shear rate shows a constant increase, with a linear behavior. Notably, at higher shear stress rates, the viscosity of the PLA-based composites tends to decrease, which is a typical behavior of a non-Newtonian pseudoplastic material. The viscosity tends to increase with the addition of the nanofiller, which is in agreement with the literature data [27,28].

### 3.3. Differential Scanning Calorimetry (DSC) Analysis

DSC measurements were carried out to investigate the influence of the HNTs on the crystallization and melting behavior of the PLA composites and their thermal properties, such as the glass transition temperature ($T_g$), crystallization temperature ($T_{cc}$), melting temperature ($T_m$), crystallization enthalpy ($\Delta H_{cc}$) melting enthalpy ($\Delta H_m$), and crystallinity ($X_c$), and the results are summarized in Table 2.

**Table 2.** DSC results of PLA and HNT compounding.

| | Material | $T_g$ (°C) | $T_{cc}$ (°C) | $\Delta H_{cc}$ (J/g) | $T_m$ (°C) | $\Delta H_m$ (J/g) | $\Delta C_p$ (J/g°C) | $X_c$ (%) |
|---|---|---|---|---|---|---|---|---|
| **1st Heating** | PLA/raw | 60.2 | 131 | 0.333 | 157.6 | 0.078 | 0.607 | 0.44 |
| | PLA | 60.2 | 132 | 16.01 | 159.6 | 34.12 | 0.397 | 53.56 |
| | PLA/HNT1 | 60.4 | 140 | 19.06 | 159.8 | 32.94 | 0.412 | 56.12 |
| | PLA/HNT3 | 61.5 | 140 | 16.26 | 160.3 | 32.32 | 0.353 | 53.51 |
| | PLA/HNT5 | 60.3 | 139 | 16.50 | 160.0 | 30.32 | 0.382 | 52.65 |
| | PLA/HNT10 | 60.8 | 140 | 14.88 | 160.7 | 31.65 | 0.325 | 55.24 |
| **2nd Heating** | PLA/raw | 59.9 | 131 | 0.518 | 157.0 | 0.59 | 0.722 | 1.18 |
| | PLA | 59.7 | 141 | 23.63 | 157.7 | 20.86 | 0.677 | 47.53 |
| | PLA/HNT1 | 60.2 | 130 | 37.51 | 157.8 | 28.40 | 0.569 | 71.13 |
| | PLA/HNT3 | 59.8 | 133 | 16.68 | 158.1 | 14.74 | 0.704 | 34.61 |
| | PLA/HNT5 | 60.2 | 134 | 7.60 | 158.1 | 7.17 | 0.580 | 16.61 |
| | PLA/HNT10 | 60.5 | 133 | 12.48 | 157.9 | 10.86 | 0.644 | 27.71 |

DSC thermographs of the PLA and PLA/HNT (1, 3, 5, 10 wt.%) are shown in Figure 3. Melting and recrystallization can occur simultaneously in the heating process; hence, the small and imperfect lamellae transform into more organized structures with a higher thermal stability [29]. PLA is a semicrystalline polymer with a very slow rate of crystallization. However, the addition of a nanofiller, such as HNT, affects its crystallization peaks due to the enhanced number of nucleation points and/or mobility of the polymeric chains. All samples containing an HNT nanofiller have a slightly higher Tm value compared to the neat PLA. The finding supports the hypothesis that HNT promotes crystallization and prevents PLA chain mobility during the melting process. However, further analyses are necessary in order to assess the significance of this change [25,30].

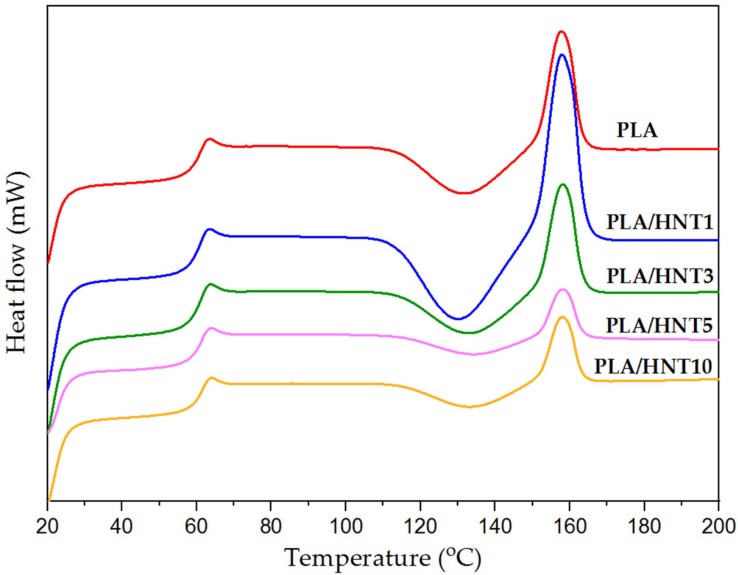

**Figure 3.** DSC thermographs of the samples of PLA and HNT with different percentages of weight concentrations.

The crystalline structure of PLA has three crystalline forms, namely $\alpha$, $\beta$, and $\gamma$. The $\alpha$ form (orthorhombic unit cell) is the most frequent and stable polymorph formed when PLA crystallizes from melt or solution. PLA has slow crystallization kinetics, and its $\alpha$ crystalline forms when cooling rates of 1–2 °C·min$^{-1}$ from the melt are applied. When the cooling rates are higher, nucleating agents are necessary in order to promote crystallization, or disordered polymeric chains are likely to be formed. Metaphase $\alpha'$ is a less stable phase and can be formed together with the $\alpha$ crystalline when the cooling rate is higher than the crystallization kinetics of the material. This $\alpha'$ phase is usually detected in PLA samples between 100 and 120 °C [31]. The addition of higher contents of HNT promotes the nucleating effect of the PLA phase, as the area corresponding to the $T_{cc}$ point slightly reduces in the case of PLA-based composites with 5 and 10% wt. HNT. [30].

### 3.4. TG/DTG (Thermal Stability Study)

The thermal stability of the PLA and PLA/HNT composites was assessed by thermogravimetric analyses (Figure 4). According to the literature, the main mass loss of neat HNT of about 10.6% in the range of 400–570 °C is related to the release of crystal water [32]. Additionally, it is known that the nanofiller HNT promotes the thermal barrier and mass transport barrier of the polymeric matrix, enhancing the thermal stability and preventing the thermal degradation of the composites. However, this effect seems to be limited to samples containing less than 10% wt. of HNT, when the thermal stability seems to have no effect on the barrier properties, as seen in Table 3. Some researchers have described this negative effect as being due to the water vapor in HNT affecting the thermal stability of the material, as the release of the adsorbed water on the surface and in the inner layers of the composite promotes the ester bond degradation of the polymeric matrix [33,34].

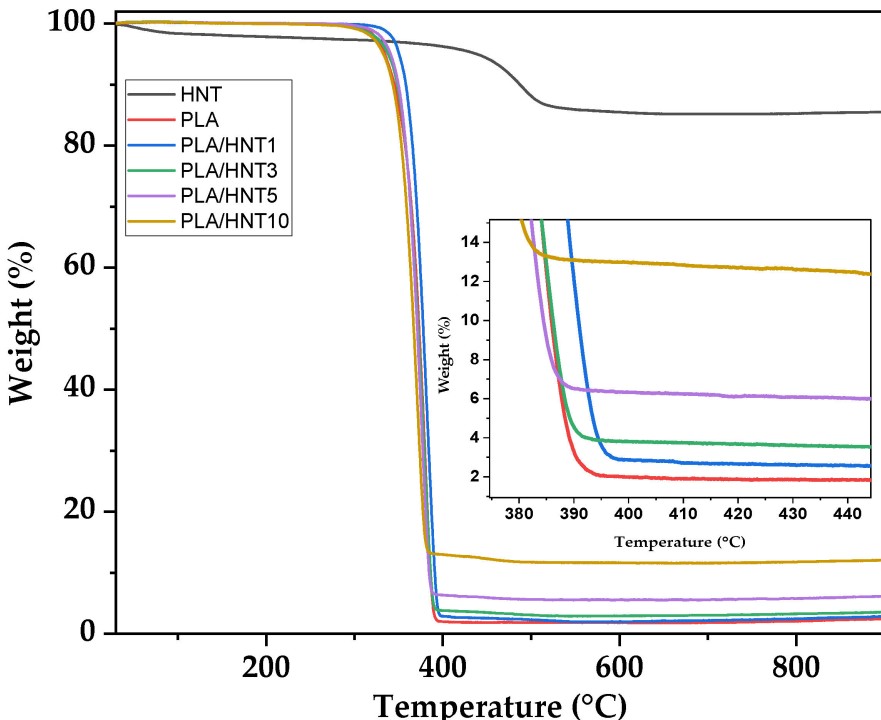

**Figure 4.** Thermogravimetric degradation of PLA samples containing 0, 1, 3, 5, and 10% wt. of HNT.

The residue content at 600 °C was measured and is summarized in Table 3. Considering the residue left by the neat PLA and the residue of HNT at the same temperature, which was 85.51%, the inorganic content of the nanocomposites was relatively close to the theoretical value of the nanofiller added during compounding, which supports the homogeneity of the clay distribution in the polymeric matrix during the material process-

ing. Similar results regarding the homogeneity of the nanofiller distribution were found in previous studies by our research group [5,25]. Figure S1 shows the TG and DTG of the PLA and PLA/HNT composites individually. All samples were confirmed to present a single thermal degradation step.

**Table 3.** TGA data of PLA and PLA/HNT composites.

| Sample | Temperature at 5% Weight Loss (°C) | Temperature at 10% Weight Loss (°C) | Temperature at Maximum Weight Loss (°C) | Residue at 600 °C (wt.%) |
|---|---|---|---|---|
| PLA | 334.2 | 345.1 | 381.1 | 1.80 |
| PLA/HNT1 | 350.4 | 358.1 | 380.8 | 1.98 |
| PLA/HNT3 | 338.1 | 348.3 | 379.3 | 2.92 |
| PLA/HNT5 | 340.8 | 349.6 | 375.4 | 5.58 |
| PLA/HNT10 | 332.7 | 343.2 | 371.2 | 11.61 |

*3.5. FTIR-ATR*

The chemical structure of PLA is $(C_3H_4O_2)_n$ and that of HNT is $Al_2Si_2O_5 \cdot (OH)_4 \cdot 2(H_2O)$. Infrared analyses were performed to assess whether molecular interactions occurred between the PLA polymeric matrix with the halloysite nanotubes after the compounding process. Figure 5 shows the infrared spectrograph in the region of 4000–450 $cm^{-1}$.

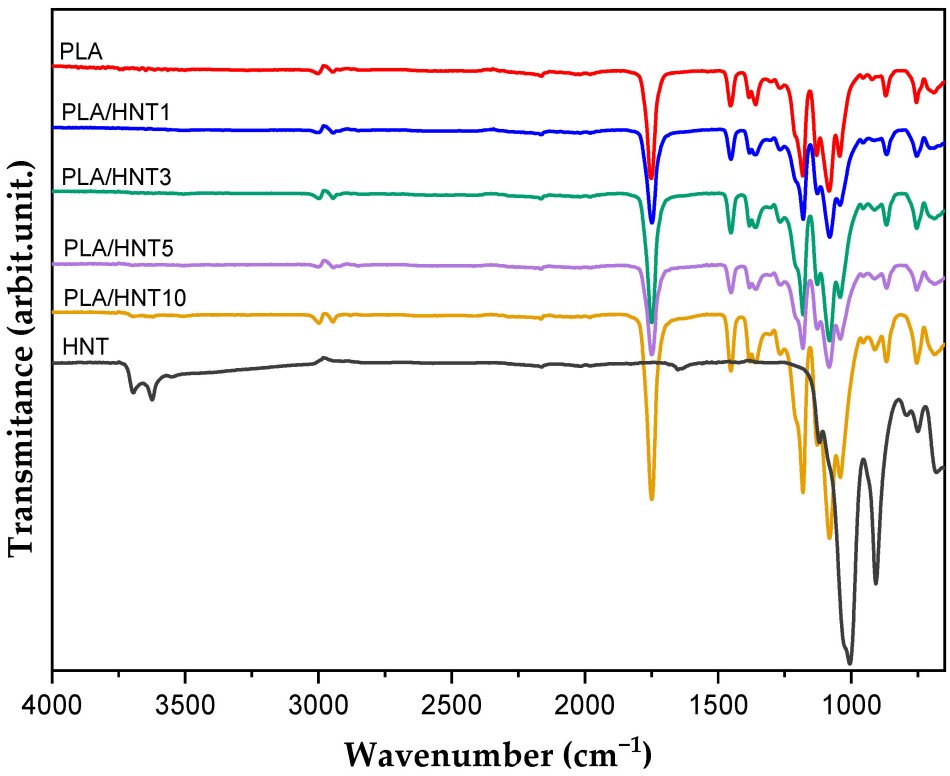

**Figure 5.** Infrared results of the material after compounding.

Neat PLA exhibited peaks at 3005 and 2955 $cm^{-1}$ can be attributed to CH group stretching, and the peaks at 1750 $cm^{-1}$ indicated –C=O carbonyl groups, while the absorption peak present at 1450 $cm^{-1}$ is attributed to the bending vibrations of $CH_3$, and the absorption bands at 1390 and 1372 $cm^{-1}$ are attributed to CH deformation and asymmetric/symmetric bending [4,25,26,35]. Previous studies have shown that the absorption peaks related to the stretching vibrations of internal and external O–H groups at 3620 and 3690 $cm^{-1}$ are associated with the absorbed and crystal water in the interlayers of HNT [29], and these absorption peaks disappear in the spectra of HNT samples dried at 500 °C in a furnace for

at least 1 h prior testing. Additionally, the spectrum of HNT displayed absorption bands at 890 cm$^{-1}$, related to Al–OH group vibrations. The spectra of the PLA/HNT composites disclosed some overlapping between the absorption bands of both PLA and HNT that are identified in the region of 1029 cm$^{-1}$ for HNT and 1044 cm$^{-1}$ for the PLA-based composites, which tended to increase the intensity of the latter as the concentration of HNT increases, as also observed in previous studies [25,26,36]. The observed sharpening of, and increase in, the intensity of the peaks are associated with the interactions between the PLA and HNT via hydrogen bonding, which may have contributed to the improvement of the thermal and mechanical properties of the PLA/HNT composites, a finding which was also supported by FTIR in previous investigations [26]. Nevertheless, the thermogravimetric analysis revealed that the crystal water adsorbed in the interlayers of HNT interferes with the thermal stability of the material, showing decomposition at lower temperatures in comparison to the neat PLA samples [25,29].

### 3.6. Wide-Angle X-Ray Diffraction—WAXD

The effect of the HNT intercalation level was detected using WAXD patterns, as seen in Figure 6. Some typical XRD peaks for the as-received HNT powders appear at 2θ = 11.90°, 20.02°, and 24.84°, in accordance with the reflection planes (001), (020), (110), and (002), with d-spacing between the crystal lattice planes of the atoms that produce the constructive interference values of 0.747, 0.450, and 0.367 nm [2,26]. The peak at 2θ = 26.68° is related to the quartz (Silica, $SiO_2$) phase of the material and is a common peak that is present in nanoclays. Impurities in halloysite usually have impacts on some applications. The use of a particular halloysite may be limited if the impurities are fine-grained and difficult to remove. In natural occurrences, free silica is usually associated with halloysite, and it was detected as a small trace component in the HNT samples, as well as a slight peak in the PLA samples containing 3, 5, and 10 wt.% of HNT [37].

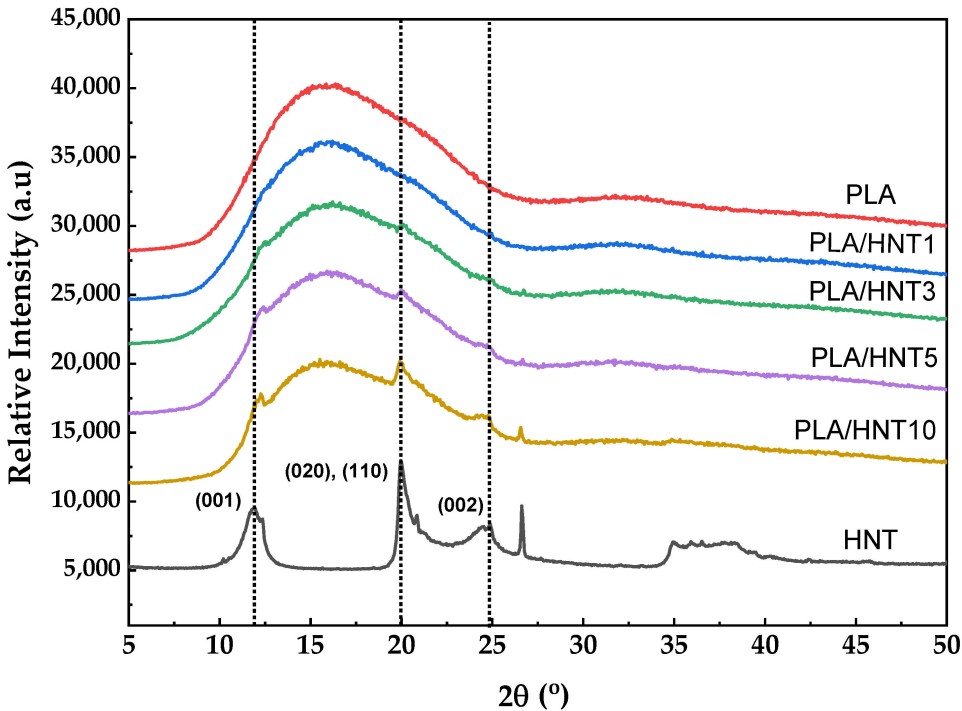

**Figure 6.** Wide-angle X-ray diffraction of PLA with different concentrations of HNT.

### 3.7. Porosity versus Infill

The relationship between the porosity and the infill percentages of the printed samples was assessed by density tests following the Archimedes principle of the density of solids when submerged in known liquids. All samples printed were theoretically set to print

with a 100% infill. However, during the porosity assessment, some samples presented with higher and lower infill values. The aim is to experimentally reach the value of a 100% infill and define the DAR which represents this value. An optical microscope is initially used to observe the DAR of the strings so as to determine the range of DAR necessary for printing cubes of 20 × 20 × 20 mm for the infill assessment, as seen in Figure 7a. For the tests using PLA, the closest value of a 100% infill is 1.52, as seen in Figure 7b. For samples of PLA/HNT5, the DAR which represents the value closest to a 100% infill is the DAR of 1.58.

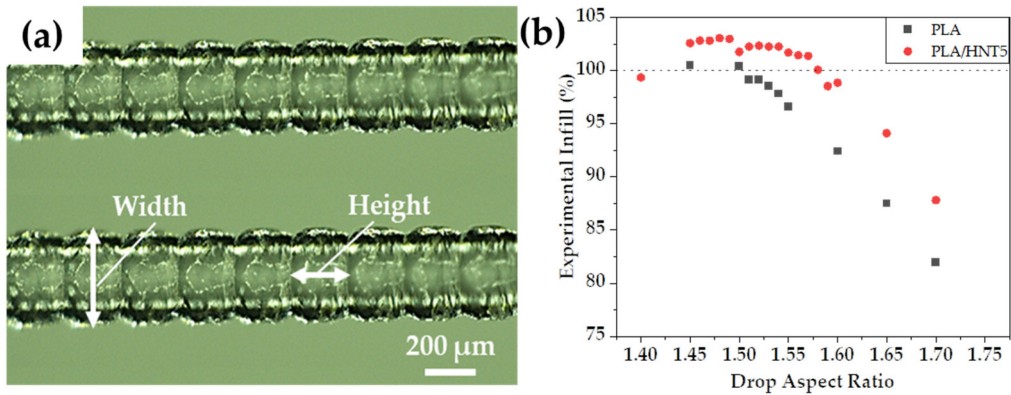

**Figure 7.** Photomicrograph of a string of printed PLA with the dimensions of the width and height of the droplet. Optical magnification of 50× in (**a**). Experimental infill percentage by drop aspect ratio (DAR = width/height) in (**b**).

The DAR is influenced by the material properties and the processing conditions [8]. In investigations using ABS, the authors of the cited study observed different optimal DAR values as the build chamber temperature varied. Nevertheless, they observed that a cooler build chamber temperature promoted the geometrical accuracy of parts; however, it may result in the poor adhesion of the strings of droplets between layers. In a similar study of the APF processing parameters using polycarbonate (PC) [13], the results revealed that DAR contributes significantly to the dimensional accuracy of printed components, and even a small increase in the DAR (i.e., theoretically, 0.2%) can increase the deposited material significantly, causing higher thermal stresses during cooling, which affect the thermal shrinkage, the DAR being more relevant than the actual droplet positioning. Additionally, the authors noticed that a 0.2% difference in the DAR values resulted in a nearly 5% variation in the final density of the printed components. These findings contribute new insights into the importance of the DAR to the body of knowledge regarding APF processing.

### 3.8. Scanning Electron Microscopy (SEM) and Energy-Dispersive X-Ray Spectroscopy (EDX)

The scanning electron microscopy photomicrographs indicated that the structure of the HNT presented with similar outer diameters of the nanotubes, ranging between 37 and 142 nm. The nanotube length was divergent and varied between 100 nm and 4.3 μm. The SEM and EDX photomicrographs can be observed in Figure 8. The material's elemental composition, measured by EDX, was recorded and found to contain only carbon and oxygen in the case of the neat PLA samples. The incorporation of HNT into the polymeric matrix resulted in the presence of microscale particles, which exhibited an elemental composition consisting of aluminum and silicon, which is indicative of HNT. Heterogeneous structures started to appear as the weight concentration of the HNT increased from 0 to 10 wt.%. Similar results were found in previous studies [38].

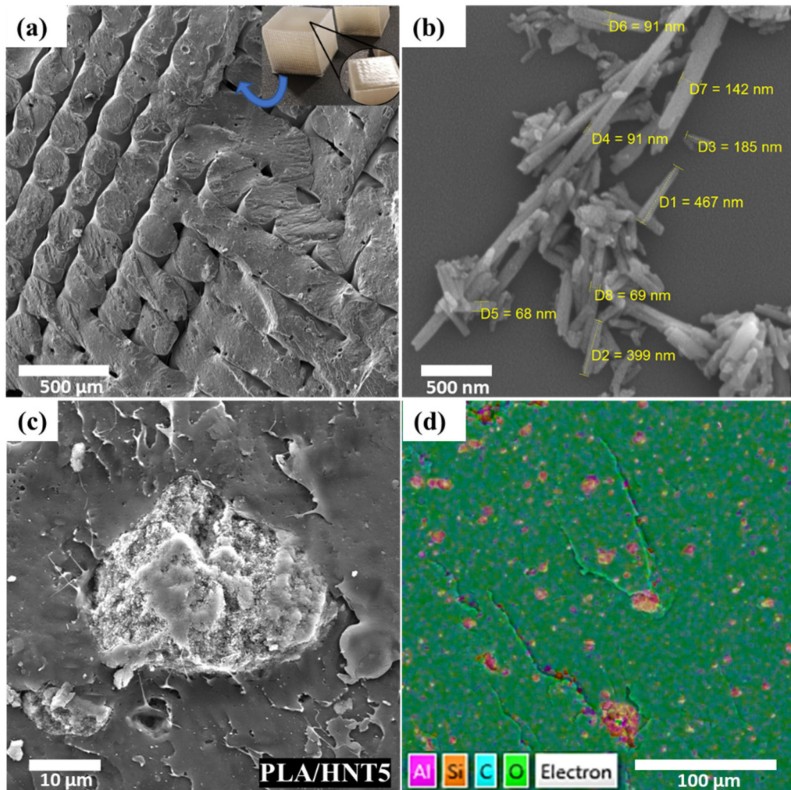

**Figure 8.** Scanning electron microscope photomicrographs: internal layers of PLA/HNT5 printed cubes after cryogenic fracture (mag 125×) in (**a**). HNT powder at a magnification of 100k× in (**b**), HNT agglomerates in PLA/HNT5 samples at a magnification of 5k× in (**c**), and EDX analysis of the PLA/HNT5 sample (mag 500×) in (**d**).

Some traces of calcium were discovered in the EDX analysis of the samples, which may be attributable to the contamination of the extruder during the compounding of the batches [25].

### 3.9. Tensile Properties

The process parameters described in Table 1 were utilized to print all the tensile specimens, which were optimized in order to obtain a good geometrical accuracy. The mechanical properties of the printed samples in the XY and XZ-orientations are shown in Figure 9, which confirms the means by which the building orientation and deposition angle affect the measured tensile properties. In the evaluated tensile tests, for instance, specimens printed in the XY direction seemed to have a higher mechanical performance than specimens printed in the XZ direction. However, the only significantly different results are those of the Young modulus and stress at break of the XY-oriented samples, which were considerably higher than those of the XZ-oriented samples. According to these findings, the samples printed in the XY-orientation are subjected to a more intense localized heating that facilitates better droplet welding, which results in an increased Young modulus and stress at break. In contrast, specimens produced in the XZ orientation have a greater surface area to volume ratio, being exposed to the build chamber's surrounding air. As a result, even if the printing path of one layer is shorter, the welding to the subsequent layer is less strong, since the temperature of the preceding layer might be lower, resulting in weaker tensile properties [8,13]. Furthermore, the deposition angle of samples starting at 0° and 45° on the first layer, with an increment of 90° on each subsequent layer, does not have considerable effects on the mechanical properties of most of the samples, except for the XY-oriented samples, which showed a drastic increase at the deposition angle of 0°.

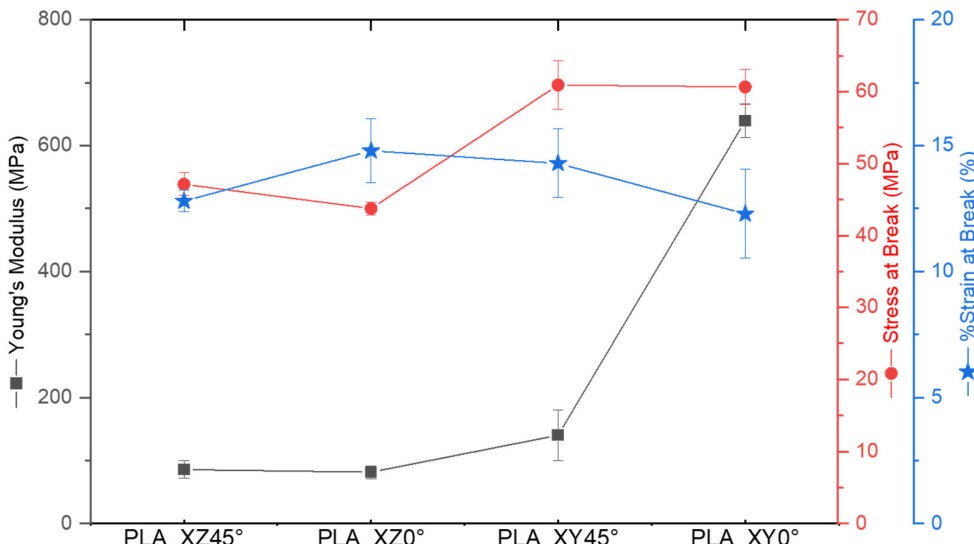

**Figure 9.** Mechanical properties of 3D-printed samples with different axis orientations (XY and XZ) and printing deposition angles: Young's modulus, stress at break, and percentage of strain at break. Force (MPa) vs. strain (mm) curves were assessed using 3D-printed samples with different axis orientations (XY and XZ) and printing deposition angles and are available in Figure S2.

The tensile readings were acquired using the equipment after each test. However, to confirm the accuracy of the data, sample data were used to calculate the Young's modulus based on the slope of the stress–strain, which yielded comparable results. When compared to different additive manufacturing techniques, namely FDM, the tensile results for the PLA samples, such as the Young's modulus, are expected to be higher, which can indicate the slippage of the samples during the tensile testing. To counteract this, samples that presented any obvious deviation from the results due to slippage were removed from the data set. To confirm this effect, a batch of samples were tested using pneumatic grips, which eliminated the possibility of slippage. The new values of the Young's modulus and stress at break found were, respectively, 2249.6 ± 23 MPa and 46.6 ± 1.5 MPa for the PLA_XY45° samples and 2923.1 ± 35 MPa and 68.5 ± 1 MPa for the PLA/HNT5_XY45° samples, which more closely reflected the values reported by the manufacturer. Nonetheless, the data presented herein remain relative. The effect of the building orientation on the APF process has already been studied in the case of ABS and poly(methyl methacrylate) (PMMA) [8,39]. The authors found out that samples printed in the XY orientation presented with significantly higher mechanical properties, such as the tensile strength, elongation at break, and the tensile modulus, when compared to specimens printed in the Z orientation (i.e., upright direction). A comparable behavior was observed in investigations of PLA in fused deposition modelling (FDM) studies [40]. Therefore, the Z orientation was not considered in this study. Additionally, the studies of the APF parameters using ABS [39] have shown that the angle of deposition which results in the highest tensile properties is 90°, which is the opposite of that which yields the maximum tensile properties in FDM processes, which is 0°. Because the deposited forms (i.e., strings of droplets for the APF and cylindrical rods for the FDM) and deposition angles are not the same, it is reasonable to predict that the results obtained for components formed by APF do not always follow the same trends as those of the components formed by FDM, which makes it challenging to compare studies using different additive manufacturing techniques. Therefore, further studies are necessary in order to compare both additive manufacturing processes [8,39].

## 4. Conclusions

Despite the emerging industrial importance of the APF technique, the current body of knowledge on this technology is still very limited, especially regarding biodegradable polymer composites, such as the PLA and HNT investigated in this study. The quality of the material is extremely important to the additive manufacturing of high-quality parts, with a high geometrical accuracy over time. Therefore, assessing the DAR of the material at certain temperatures is essential. The determined DAR values for the PLA and PLA/HNT5 composites produced high-quality samples, which allowed us to determine the tensile properties of tensile specimens printed with different axis orientations and deposition angles, the samples of PLA being printed in the XY axis direction with a deposition angle starting at 0°, the value yielding the highest mechanical properties (Young modulus and stress at break). Tests with higher concentrations of HNT in the PLA matrix (up to 10% wt. of HNT) resulted in the clogging of the nozzle and the impossibility of producing a high-quality printed component due to the higher content of nanoparticles agglomerating at the nozzle end. Therefore, no results were obtained using higher concentrations of HNT. The current methodology achieved satisfactory results with respect to the 3D printing and evaluation of successful PLA/HNT composites using the APF technique for the first time. These composites can serve as feedstock for the future additive manufacturing of medical devices (namely, biodegradable ureteral stents). Additionally, APF was proven to produce comparable results to standard manufacturing techniques, such as injection molding and other 3D printing methods.

**Supplementary Materials:** The following supporting information can be downloaded at: https://www.mdpi.com/article/10.3390/jcs6100309/s1, Figure S1: TG/DTG curves of PLA and PLA/HNT composites. PLA in (**a**), PLA/HNT1 in (**b**), PLA/HNT3 in (**c**), PLA/HNT 5 in (**d**), and PLA/HNT10 in (**e**).; Figure S2: Force (MPa) vs. strain (mm) curves were assessed from 3D-printed samples with different axis orientations (XY and XZ) and printing deposition angles. PLA_XY0° in (**a**), PLA_XY45° in (**b**), PLA_XZ0° in (**c**), and PLA_XZ45° in (**d**).

**Author Contributions:** Conceptualization, D.M.D. and L.G.E.; methodology, L.G.E.; investigation, L.G.E.; writing—original draft preparation, L.G.E.; writing—review and editing, J.S.C. and D.M.D.; supervision, J.S.C.; I.M.; N.M.G. and D.M.D.; project administration, D.M.D.; funding acquisition, N.M.G., I.M., and D.M.D. All authors have read and agreed to the published version of the manuscript.

**Funding:** This research was funded by Enterprise Ireland under the capital call of 2019, grant number CE 20190068, the Applied Polymer Technology Gateway Project No. TG-2017-1014, and the AIT President's Seed Fund (PSF2020-DD).

**Data Availability Statement:** Not applicable.

**Acknowledgments:** The authors would like to acknowledge the technician Rodrigo A. Barbieri from the University of Caxias do Sul for his support with WAXD and SEM analyses.

**Conflicts of Interest:** The authors declare no conflict of interest.

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
