# Peer review of "Process Optimization for the 3D Printing of PLA and HNT Composites with Arburg Plastic Freeforming"

_jcs, doi:10.3390/jcs6100309_

Round 1

Reviewer 1 Report

The paper entitled "Process optimization for the 3D printing of PLA and HNT composites with the Arburg Plastic Freeforming" should be published after some minor corrections.

The quality of the manuscript preparation is very high. Since the use of Freeformed printed is not common, the novelty of the presented results is  also high. 

I have no objections to any of the key aspects of the work, i.e. the methodology of work, the presentation of results and discussion. They are presented with a lot of details. The only remark refers to the results of the tensile tests, since the modulus values are relatively low, considering the use of PLA. Even for FDM printed samples the stiffness of samples can reach 2 or 3 GPa, while here the results indicate miaximum 0.6 GPa. The rather unusual appearance of the tensile curves may suggest that the sample is slipping out of the machine jaws, which indirectly also affects the rather high elongation at break.

Reviewer 2 Report

The paper entitled “Process optimization for the 3D printing of PLA and HNT composites with the Arburg Plastic Freeforming” described the PLA/HNT composites prepared by 3D printing. The biodegradable PLA/HNT polymer composite will be attractive for many readers. The experimental data is well supported the conclusion. I suggest it to be accepted by this journal after revising the following comments.

1.       Line 279, please state how many specimen you have tested and what is the dimension of this samples in the tensile tests.

2.       Line 398, please highlight the scale bars in the SEM images in figure 8.

3.       Please adjust the right Y-axis scale so that figure 9 can be observed more clearly.

4.       The conclusion part should be rewritten. The authors should focus on the effects of the 3D printing and the HNT content on the composites.

5.       Please revise the references carefully, and follow the reference format of this journal.

6.       There are some grammar mistakes. Please revise them carefully.

Round 2

Reviewer 1 Report

I have no other comments. 

Congratulations on your interesting publication.

Reviewer 2 Report

The authors have answered the comments well. I recommend it to be published on this journal.